# Synthesis and Structural Studies of *peri*-Substituted Acenaphthenes with Tertiary Phosphine and Stibine Groups [note 1]

**DOI:** 10.3390/molecules29081841

**Published:** 2024-04-18

**Authors:** Laurence J. Taylor, Emma E. Lawson, David B. Cordes, Kasun S. Athukorala Arachchige, Alexandra M. Z. Slawin, Brian A. Chalmers, Petr Kilian

**Affiliations:** 1School of Chemistry, University of Nottingham, Nottingham NG7 2RD, UK; 2EaStCHEM School of Chemistry, University of St Andrews, North Haugh, St Andrews, Fife KY16 9ST, UK; 3Centre for Microscopy and Microanalysis, The University of Queensland, St Lucia, QLD 4072, Australia

**Keywords:** *peri*-substitution, phosphorus, antimony, NMR, single-crystal X-ray structures, synthesis, QTAIM, EDA-NOCV, pnictogen bond

## Abstract

Two mixed *peri*-substituted phosphine-chlorostibines, Acenap(P*i*Pr_2_)(SbPhCl) and Acenap(P*i*Pr_2_)(SbCl_2_) (Acenap = acenaphthene-5,6-diyl) reacted cleanly with Grignard reagents or *n*BuLi to give the corresponding tertiary phosphine-stibines Acenap(P*i*Pr_2_)(SbRR’) (R, R’ = Me, *i*Pr, *n*Bu, Ph). In addition, the Pt(II) complex of the tertiary phosphine-stibine Acenap(P*i*Pr_2_)(SbPh_2_) as well as the Mo(0) complex of Acenap(P*i*Pr_2_)(SbMePh) were synthesised and characterised. Two of the phosphine-stibines and the two metal complexes were characterised by single-crystal X-ray diffraction. The *peri*-substituted species act as bidentate ligands through both P and Sb atoms, forming rather short Sb-metal bonds. The tertiary phosphine-stibines display through-space *J*(CP) couplings between the phosphorus atom and carbon atoms bonded directly to the Sb atom of up to 40 Hz. The sequestration of the P and Sb lone pairs results in much smaller corresponding *J*(CP) being observed in the metal complexes. QTAIM (Quantum Theory of Atoms in Molecules) and EDA-NOCV (Energy Decomposition Analysis employing Naturalised Orbitals for Chemical Valence) computational techniques were used to provide additional insight into a weak n(P)→σ*(Sb-C) intramolecular bonding interaction (pnictogen bond) in the phosphine-stibines.

## 1. Introduction

Tertiary amines and phosphines play a key role as tuneable ligands, with uses in transition metal catalysis and other applications. The heavier tertiary pnictines (ER_3_, E = As, Sb, Bi, R = alkyl, aryl) also serve as L-type ligands in a number of complexes, although they generally display lower donor strength than corresponding N and P-based ligands [1,2].

Several unusual properties stemming from the close-proximity of two pnictine groups in *peri*-substituted scaffolds have been noted. The first of these was in the 1960’s, when the remarkably high basicity of proton sponge 1,8-bis(dimethylamino)naphthalene (**A**, Figure 1) was reported by Alder [3]. Tertiary bis(phosphines) such as the phosphorus analogues of the proton sponge **B** (Figure 1) were reported shortly thereafter [4,5], as were several of their metal complexes [6,7,8].

Syntheses of *peri*-substituted tertiary bis(arsines), such as **C** (Figure 1), and their complexes were also reported as early as the 1960’s [9]. However, no crystal structures of such ligands or complexes have appeared in the literature. Prototypical naphthalene bis(stibines) with dimethylstibino (**D1**) and diphenylstibino groups (**D2**) were synthesised by Reid, together with their Mo(0) and Pt(II) complexes [10]. The aryl species **D2** (both naphthalene and acenaphthene variants) and **D3** (naphthalene variant) were recently structurally characterised by Schulz, together with the two bis(bismuthines) **E** [11,12]. However, no structural data for any of the bis(stibine) or bis(bismuthine) metal complexes have been published to date.

The related Sb−Sb bonded species **F** [12], as well as the doubly backboned species **G** [13,14,15] and **H** [11], have received significant attention recently, and a few transition metal complexes with these as ligands have also been reported [13].

Species **I** (Figure 1) with two differing Group 15 *peri*-atoms display intriguing dative bonding and NMR properties [16,17]. Surprisingly, only two bis(tertiary) phosphine-stibine and phosphine-bismuthine *peri*-substituted species have been reported to date: **I_Sb_** [18] and **I_Bi_** [19]. Both of these display repulsive interactions between the two pnictogen-centred groups, although their geometries indicate a weak pnictogen bond (nP→σ*(Pn-C)) is present, as indicated in Figure 1 by a dashed line. This is in contrast to the related E(III)−E(III) halophosphines, such as **J** [18,20] and **K** [18,21], which display strong dative pnictogen-pnictogen bonds.

Apart from the phosphine-stibine **I_Sb_** [18], the most closely related work to this paper are the geminally substituted bis- and tris(acenaphthene) species **L** and **M** [22,23]. Only one metal complex of these has been structurally characterised, the Rh(I) species **N** [23].

As a continuation of our synthetic, structural and bonding studies of *peri*-substituted species, we investigated the utility of the halostibines **J** and **K** (Figure 1), reported by us earlier [18], as synthons towards primary stibine functionalities. Prompted by the paucity of the literature data, we have also probed the coordination chemistry of the produced tertiary phosphine-stibines.

## 2. Results and Discussion

### 2.1. Synthesis and Spectroscopic Properties of the Tertiary Stibines ***4***, ***5***, ***7***, and ***8***

The three major precursors for the syntheses reported in this paper were bis(aryl) stibine **2**, chloro(aryl) stibine **3** and dichlorostibine **6** (Figure 1). Syntheses and structural information for these compounds, starting from **1**, have been reported by us [18].

The reactive Sb−Cl motifs in chlorostibine **3** and dichlorostibine **6** were used to form new Sb−C bonds via reactions with carbon nucleophiles. The chlorostibine **3** was reacted with one equivalent of alkyl-Grignard reagents, MeMgBr and *i*PrMgCl, to afford alkyl-aryl stibines **4** (86%) and **5** (81%), respectively.

The reaction of dichlorostibine **6** with two equivalents of MeMgBr afforded dimethylstibine **7** as an off-white solid (yield ca. 90%; exact yield determination was not possible as ^1^H NMR indicated solvation by Et_2_O). Reacting *n*BuLi with **6** also resulted in Sb−C bond formation; reaction with two equivalents of *n*BuLi afforded crude di-*n*-butylstibine **8** in quantitative yield (obtained as an oil). The crystallisation of **8** from common organic solvents was not successful. However, a small amount of crystalline **8** was obtained through the long standing of the oil at room temperature (see below).

All the Sb−C bond-forming reactions were remarkably clean, as judged by ^31^P{^1^H} and ^1^H NMR spectroscopy. The newly prepared compounds were further characterised by ^13^C DEPT-Q NMR, HRMS (peaks corresponding to (M + H)^+^ with correct isotopic patterns were observed in all cases) and (for **7** and **8**) also by Raman spectroscopy. The purity of **5** was confirmed by CHN microanalysis. The novel tertiary stibines appear to be hydrolytically stable (in some cases, an aqueous wash was involved in the work-up); however they are oxidised in the presence of air. Both **7** and **8** decomposed in chloroform solutions within several days, indicating instability in halogenated solvents.

The reaction of **6** with one equivalent of *n*BuLi gave an oil after the workup. This oil was shown by ^31^P{^1^H} NMR to be a complex mixture, with a major peak at δ_P_ 18.5 ppm, corresponding to the doubly substituted species **8**, indicating that selective single substitution using an organolithium as a nucleophile may be difficult to achieve.

The ^31^P{^1^H} NMR spectra of the phosphine-stibines **4**, **5**, **7** and **8** display singlets within a narrow range of δ_P_ (−18.5 to −20.8 ppm). Notable through-space couplings (indicated by the TS superscript in the *J* notation, ^TS^*J*) are observed in the ^13^C{^1^H} NMR spectra between the phosphorus atom and carbons attached to the antimony atom. In **4**, the *ipso*-C of the Sb-Ph moiety shows a ^5TS^*J*_CP_ of 16.2 Hz. An even larger ^5TS^*J*_CP_ of 34.1 Hz is observed for the Sb-CH_3_ of **4**. Interestingly, the acenaphthene *ipso*-carbon atom shows no detectable coupling to the phosphorus atom, despite having a shorter bond path (formally ^3^*J*_CP_).

A similar situation is observed in **5** (^5TS^*J*_CP_ = 17.3 Hz (*ipso*-Ph) and ^5TS^*J*_CP_ = 36.8 Hz (Sb-CH), although in this case small magnitude splitting with the *ipso*-acenaphthene carbon (C1 in the numbering scheme shown in Figure 2) is observable (^3^*J*_CP_ = 2.4 Hz). Similar magnitudes of *J*_CP_ involving carbon atoms bonded directly to Sb atoms are also observed for **7** (^5TS^*J*_CP_ = 34.9 Hz (CH_3_); ^3^*J*_CP_ = 5.1 Hz (C1, Acenap)) and **8** (^5TS^*J*_CP_ = 30.6 Hz (CH_2_); ^3^*J*_CP_ = 5.0 Hz (C1, Acenap)). Observation of the through-space couplings in **4**, **5**, **7** and **8** is consistent with the significant overlap of P and Sb lone pairs as confirmed by single crystal X-ray diffraction (*vide infra*) and is in agreement with observations made in our previous study of P-Sb acenaphthenes [18].

### 2.2. Synthesis and Spectroscopic Properties of Tertiary Stibine Metal Complexes ***2.PtCl_2_*** and ***4.Mo(CO)_4_***

*Peri*-substituted species **2** and **4**, bearing tertiary phosphine and tertiary stibine groups, were reacted with platinum(II) and molybdenum(0) motifs to explore their coordination chemistry. It was of interest to see if the phosphine-stibine species would act as bidentate ligands, with the metal coordinating through both phosphorus and antimony atoms.

[PtCl_2_(cod)] was reacted with **2** in dichloromethane, giving **2.PtCl_2_** as a yellow powder in a good yield (76%). Similarly, the reaction of [Mo(CO)_4_(nbd)] with **4** in dichloromethane gave **4.Mo(CO)_4_** as a brown powder in a near-quantitative yield (Figure 1). Both complexes were stable to air in the solid and solution in the chlorinated solvents used to acquire their NMR spectra (CD_2_Cl_2_ and CDCl_3_, respectively).

The ^31^P{^1^H} NMR spectrum of **2.PtCl_2_** consists of a singlet with a set of ^195^Pt satellites (δ_P_ 7.8 ppm, ^1^*J*_PPt_ = 3357 Hz), with the complementary doublet observed in the ^195^Pt{^1^H} NMR spectrum (δ_Pt_ −4541 ppm). The coordination of platinum centres resulted in a high-frequency shift (c.f. free ligand **2**, δ_P_ −21.9 ppm) [18] as well as loss of the through-space *J*_CP_ coupling (c.f. ^5TS^*J*_CP_ 40.3 Hz for *ipso*-Ph carbon in **2**).

Coordination of the Mo(CO)_4_ fragment to **4** resulted in an even more pronounced high-frequency shift for **4.Mo(CO)_4_** (*δ*_P_ 43.4; c.f. *δ*_P_ −19.6 ppm in free ligand **4**). Similar to **2.PtCl_2_**, the *J*_CP_ couplings between the phosphorus atom and the carbon atoms adjacent to the antimony atom are much smaller magnitudein **4.Mo(CO)_4_** than **4**. This is notable as the through-bond coupling paths are shorter in the complex (formally ^3^*J*_CP_, 2.3 and 2.9 Hz), compared to those in the free ligand **4** (^5TS^*J*_CP_, 16.2 and 34.1 Hz).

### 2.3. Structural Discussion

Two of the phosphine-stibines (**5** and **8**), as well as the two complexes **2.PtCl_2_** and **4.Mo(CO)_4_**, were subjected to the single crystal diffraction study. The structures are shown in Figure 2 and Figure 3 and Table 1.

Crystals of **5** were grown from ethanol. The structure of **5** displays a moderately strained geometry, with a P9∙∙∙Sb1 distance of 3.172(3) Å (129% of ∑r_covalent_, 76% of ∑r_vdW_) [24,25] and a splay angle of 15.1(12)°. These parameters indicate that, while the two functional groups in the *peri*-positions are forced into close proximity, the P∙∙∙Sb interaction is primarily repulsive. However, a more detailed look at the *peri*-region geometry indicates the presence of a weak intramolecular pnictogen bond (n(P)→σ*(Sb–C_iPr_)), which manifests through a quasi-linear arrangement of the P9∙∙∙Sb1−C19 motif (168.5°, see Figure 3).

Crystals of **8** were obtained by prolonged standing of the crude oily product. The molecule of **8** in the structure displays a similar geometry to **5**, with a slightly larger P∙∙∙Sb distance of 3.218(2) Å. In contrast to **5**, the “homoleptic” substitution pattern of the Sb atom in **8** allows direct comparison of the Sb-C bond lengths for the two *n*-butyl groups. This reveals that the Sb1−C13 bond length is significantly elongated compared to the Sb1−C17 bond length (2.197(6) vs. 2.100(7) Å). This indicates the donation of electron density (n(P9)) into the (antibonding) σ*(Sb1−C13) orbital, consistent with the formation of a (quasi-linear) pnictogen bond n(P9)→σ*(Sb1–C3), P9···Sb1−C13 angle 167.1°, see Figure 3.

Crystals of **2.PtCl_2_** were grown from dichloromethane/hexane with a solvated molecule of dichloromethane. The platinum atom adopts a distorted square planar geometry, with the P and Sb atoms of ligand **2** bound in a *cis* fashion. Coordination of the PtCl_2_ fragment results in elongation of the P∙∙∙Sb distance to 3.357(12) Å (c.f. 3.191(1) Å in **2**) and significantly increased out-of-plane distortions within the acenaphthene ligand (see Table 1) [18]. While the P−Pt distance (2.248(4) Å) is as expected, the Sb−Pt distance in **2.PtCl_2_** (2.4570(10) Å) is one of the shortest Sb–Pt bonds known, most likely due to the geometric constraints of the ligand. Of the 143 Sb–Pt bonds recorded in the Cambridge Structural Database to date, only 6 are shorter than the bond in compound **2.PtCl_2_**. Those 6 examples are all Sb(V) species with highly electrophilic Sb centres [23,26,27,28], hence **2.PtCl_2_** is the shortest Pt−Sb bond for a stibine (R_3_Sb) ligand.

Crystals of **4.Mo(CO)_4_** were grown from hexane. The molybdenum adopts a (distorted) octahedral geometry as expected, with ligand **4** attached in *cis* fashion. As above, the P-Mo distance is as expected; however, the Sb–Mo bond length of 2.7007(6) Å, is one of the shortest Sb–Mo bonds known. Of the 518 independent Sb–Mo bonds (in 97 compounds) recorded in the Cambridge Structural Database, only 8 are shorter than the bond in compound **4.Mo(CO)_4_**. The three compounds showing the shortest distances (the shortest being 2.64386(19) Å) are all stibine (or halostibine) complexes, possessing a tridentate scaffold combining stibine and phosphine functionalities, with similar constraints as those seen in **4** [29].

### 2.4. Computational Analysis

DFT calculations were employed to further investigate the nP→σ*(Sb–C) interaction in **5** and **8**. Geometry optimisations were performed on these compounds (PBE0/SARC-ZORA-TZVP for Sb, PBE0/ZORA-def2-TZVP for all other atoms), and the resulting structures were in good agreement with the X-ray geometries. In particular, the Sb−C bond lengths in **8** were well reproduced, with the Sb−C bond opposite the P atom being elongated (Sb1−C13 2.197(6) Å experimental, 2.208 Å calculated; Sb1−C17 2.100(7) Å experimental, 2.175 Å calculated).

A Quantum Theory of Atoms in Molecules (QTAIM) [30,31] analysis was applied to **5** and **8.** Bond critical points (BCPs) were located between the Sb1 and P9 atoms for both **5** and **8**, indicative of a bonding interaction. Selected QTAIM parameters evaluated at BCPs for these molecules are summarized in Table 2. The Sb1···P9 BCPs all display a relatively low electron density (ρ_BCP_) and a small and positive Laplacian (∇^2^ρ_BCP_), which are typical of interactions between heavier elements [32,33].

The bond degree parameter [32] (BD = H_BCP_/ρ_BCP_; H_BCP_ = energy density at BCP) [32,34] and the ratio of |V_BCP_|/G_BCP_ [32] (V_BCP_ = electronic potential energy at the BCP and G_BCP_ = electronic kinetic energy at the BCP) are two valuable metrics in QTAIM for analysing bonds between heavier elements. The BD indicates the amount of covalency in a bond, with larger negative values denoting a greater covalent interaction [32,34]. The P9∙∙∙Sb1 interactions in **5** and **8** both show small, negative values, suggesting a weakly covalent interaction (Table 2). |BD| is smaller for **8** than **5**, suggesting less covalency in the P9∙∙∙Sb1 interaction for **8**. This can be rationalised by the Sb(*n*Bu)_2_ moiety being more electron-rich than Sb(Ph)*i*Pr, and thus a poorer electron acceptor. The interaction energy (E_i_ = $\frac{1}{2}V_{BCP}$) [35], which can be used as a rough estimate of bond strength, similarly indicates a weaker P9∙∙∙Sb1 interaction in **8**.

The |V_BCP_|/G_BCP_ ratio differentiates between different bond types: purely closed-shell interactions such as van der Waals or ionic bonds exhibit |V_BCP_|/G_BCP_ < 1, while fully covalent interactions show |V_BCP_|/G_BCP_ > 2. Bonds with intermediate ratios (1 < |V_BCP_|/G_BCP_ < 2) are termed transit closed-shell interactions, such bonds possess partial covalent character [32]. Both **5** and **8** display 1 < |V_BCP_|/G_BCP_ < 2, with a larger value for **5** than **8**. This once again suggests a more covalent P9∙∙∙Sb1 in **5** than **8**. Crucially, the BD and |V_BCP_|/G_BCP_ suggest that the P9∙∙∙Sb1 interaction in both **5** and **8** is not purely closed shell (i.e., Van der Waals), and that there is some degree of electron sharing between the P and Sb atoms, consistent with a nP→σ*(Sb–C) interaction. Also of note is the difference in QTAIM parameters for Sb1−C13 and Sb1−C17 in **8**. Sb−C13 shows a slightly reduced BD, |V_BCP_|/G_BCP_ and E_i_ compared with Sb−C17 (Table 2), consistent with a weakening of the Sb1–C13 bond due to donation into the Sb–C σ* orbital.

This P9∙∙∙Sb1 interaction was further probed by an Energy Decomposition Analysis employing Naturalised Orbitals for Chemical Valence (EDA-NOCV) [36,37,38,39]. This allows the donor-acceptor interaction between P9 and Sb1 to be visualised and also allows for quantification of the interaction energy. For this analysis, the molecules were divided into two closed-shell fragments: an Acenap(P*i*Pr_2_)^−^ anion and an SbR_2_^+^ cation. The total interaction energy between these fragments (ΔE_int_) was computed and divided into terms for ΔE_steric_, ΔE_orb_ and ΔE_disp_ (Table 3). ΔE_orb_ and ΔE_disp_ are the orbital and dispersion interaction energies, respectively. ΔE_steric_ is the combined electrostatic attraction and Pauli-repulsion energy terms [40]. In both **5** and **8**, ΔE_steric_ is negative, indicating a significant electrostatic attraction. This is a result of formally assigning the fragments as cationic and anionic. **8** is observed to have a slightly smaller ΔE_int_, ΔE_steric_, ΔE_orb_ and ΔE_disp_ than **5**, which can again be contributed to more electron rich groups on Sb weakening the donor-acceptor interaction.

The ΔE_orb_ term can be broken down into pairs of natural orbitals for chemical valence (NOCVs), which represent the orbital interactions between the Acenap(P*i*Pr_2_)^−^ and SbR_2_^+^ fragments. For each pair of NOCVs, a deformation density plot (Δ*ρ*_k_), which represents the flow of electrons between the molecular fragments, and its corresponding energy contribution to ΔE_orb_, can be determined [36]. The first deformation density plots (Δ*ρ*_1_) for **5** and **8**, which have the largest energetic contribution to ΔE_orb_, are dominated by electron flow from the (anionic) carbon of Acenap(P*i*Pr_2_)^−^ to the Sb atom. However, Δ*ρ*_2_ and Δ*ρ*_3_ for **5** and **8** both appear to show electron donation from the P lone pair to a Sb−C σ* orbital (Figure 4). The energy contributions of these interactions are Δ*ρ*_2_ = −15.8 kcal mol^−1^, Δ*ρ*_3_ = −12.1 kcal mol^−1^ for **5**; Δ*ρ*_2_ = −14.4 kcal mol^−1^, Δ*ρ*_3_ = −11.0 kcal mol^−1^ for **8**. These values are likely a significant overestimate of the nP→σ*(Sb–C) interaction energy, as the deformation density plots also show significant contributions from the π-systems of **5** and **8**. However, these plots do strongly support the existence of donor-acceptor interactions between P9 and Sb1 in both compounds. Note that blue isosurface (an indicator of accepting electron density) is primarily observed on the Sb–C bond *opposite* the P-atom and not the other Sb–Ph (**5**) or Sb–*n*Bu (**8**) bond (Figure 4).

## 3. Experimental Section

### 3.1. General Considerations

Unless otherwise stated, all experimental procedures were carried out under an atmosphere of dry nitrogen using standard Schlenk techniques or under an argon atmosphere in a Saffron glove box. Dry solvents were used unless otherwise stated and were either collected from an MBraun SPS-800 Solvent Purification System, or dried and stored according to literature procedures [41]. The peri-substituted acenapthene precursors **1** [42], **2**, **3** and **6** [18] were synthesised according to literature procedures. “*In vacuo*” refers to a pressure of ca. 2 × 10^−2^ mbar.

### 3.2. NMR Spectroscopy

All novel compounds were characterised where possible by ^1^H, ^13^C DEPTQ and ^31^P{^1^H} NMR spectroscopy, including measurements of ^1^H{^31^P}, H-H DQF COSY, H-C HSQC, H-C HMBC and H-P HMBC. ^13^C{^1^H} NMR spectra were recorded using the DEPTQ-135 pulse sequence with broadband proton decoupling. Measurements were performed at 20 °C using a Bruker Avance 300, Bruker Avance II 400 or Bruker Avance III 500 (MHz) spectrometer. For both ^1^H and ^13^C NMR, chemical shifts are relative to Me_4_Si, which was used as an external standard. The residual solvent peaks were used for calibration (CHCl_3_, δ_H_ 7.26, δ_C_ 77.16 ppm; CD_2_Cl_2_, δ_H_ 5.32, δ_C_ 53.84 ppm). For ^31^P NMR, 85% H_3_PO_4_ in D_2_O (δ_P_ 0 ppm) was used as an external standard. ^195^Pt NMR was acquired for **2.PtCl_2_**, and 1.2 M Na_2_[PtCl_6_] in D_2_O (δ_Pt_ 0 ppm) was used as the external standard. The NMR numbering scheme is shown in Figure 5.

### 3.3. Other Analyses

Elemental analyses (C, H and N) were performed at London Metropolitan University. High resolution mass spectrometry was performed by the EPSRC UK National Mass Spectrometry Facility (NMSF) at Swansea University using either a Waters Xevo G2-S (ASAP) or a Thermofisher LTQ Orbitrap XL (APCI) mass spectrometer. Electrospray ionisation (ES) spectra were acquired at the University of St Andrews Mass Spectrometry Facility using a Thermo Exactive Orbitrap Mass Spectrometer. Both IR and Raman spectra were collected on a Perkin Elmer 2000 NIR FT spectrometer. KBr tablets were used in IR measurements; powders in sealed glass capillaries were used for Raman spectra acquisitions. Melting (or decomposition) points were determined by heating solid samples in glass capillaries using a Stuart SMP30 melting point apparatus.

### 3.4. [iPr_2_P-Ace-SbPh_2_]PtCl_2_, ***2.PtCl_2_***

To a suspension of dichloro(1,5-cyclooctadiene)platinum(II) (72 mg, 170 µmol) in dichloromethane (4 mL), a solution of **2** (100 mg, 170 µmol) in dichloromethane (10 mL) was added dropwise. The solution was left to stir at room temperature overnight. The volatiles were removed *in vacuo* to give **2.PtCl_2_** as a pale-yellow powder (104 mg, 76%). Crystals suitable for X-ray diffraction were grown from the vapour diffusion of hexane into a saturated solution of the compound in dichloromethane. M.p. 108 °C with decomposition.

**^1^H NMR:** δ_H_ (500.1 MHz, CD_2_Cl_2_) 8.09 (1H, dd, ^3^*J*_HP_ 11.0, ^3^*J*_HH_ 7.6 Hz, H-8), 7.71 (1H, d, ^3^*J*_HH_ 7.0 Hz, H-2), 7.67–7.63 (4H, m, *o*-Ph CH), 7.57 (1H, d, ^3^*J*_HH_ 7.5 Hz, H-7), 7.54–7.51 (2H, m, *p*-Ph CH), 7.49–7.44 (5H, m, H-3, *m*-Ph CH), 3.55–3.45 (6H, m, H-11, 12, 2× *i*Pr CH), 1.36 (6H, dd, ^3^*J*_HP_ 18.2, ^3^*J*_HH_ 7.0 Hz, 2× *i*Pr CH_3_), 1.17 (6H, dd, ^3^*J*_HP_ 16.1, ^3^*J*_HH_ 7.0 Hz, 2× *i*Pr CH_3_).

**^13^C DEPTQ NMR:** δ_C_ (125.8 MHz, CD_2_Cl_2_) 152.9 (s, qC-6), 152.7 (s, qC-4), 140.3 (d, ^3^*J*_CP_ 7.2 Hz, qC-5), 139.8 (d, ^2^*J*_CP_ 6.8 Hz, qC-10), 139.1 (s, C-2), 135.7 (d, ^2^*J*_CP_ 4.3 Hz, C-8), 135.5 (s, *o*-Ph CH), 131.1 (s, *m*-Ph CH), 129.4 (s, *p*-Ph CH), 126.6 (s, *i*-Ph qC), 120.4 (s, C-3), 119.4 (d, ^3^*J*_CP_ 9.4 Hz, C-7), 114.0 (d, ^1^*J*_CP_ 48.4 Hz, qC-9), 111.2 (d, ^3^*J*_CP_ 7.8 Hz, qC-1), 30.5 (s, C-11/12), 30.0 (s, C-11/12), 29.9 (d, ^1^*J*_CP_ 34.7 Hz, 2× *i*Pr CH), 19.4 (s, 2× *i*Pr CH_3_), 19.3 (s, 2 × *i*Pr CH_3_).

**^31^P{^1^H} NMR:** δ_P_ (202.5 MHz, CD_2_Cl_2_) 7.8 (s with ^195^Pt satellites, ^1^*J*_PPt_ 3357.0 Hz).

**^195^Pt{^1^H} NMR:** δ_Pt_ (107.0 MHz, CD_2_Cl_2_) −4541 (d, ^1^*J*_PtP_ 3357.0 Hz).

**IR** (KBr disc, cm^−1^) ν_max_ 3047m (ν_Ar–H_), 2925s (ν_C–H_), 1601s, 1479m, 1434vs, 1334m, 1254m, 1033m, 998w, 848m, 734vs, 693s, 451m, 270m, 241s.

**Raman** (glass capillary, cm^−1^) *ν*_max_ 3052s (ν_Ar–H_), 2926s (ν_C–H_), 1604m, 1578m, 1444m, 1337m, 1000vs, 661s, 581m, 319s, 180vs.

**MS** (ES+): *m*/*z* (%) 775.06 (100) [M − Cl].

### 3.5. iPr_2_P-Ace-Sb(Ph)Me, ***4***

To a stirred suspension of **3** (0.50 g, 0.99 mmol) in tetrahydrofuran (20 mL), cooled to −78 °C, methylmagnesium bromide (0.50 mL, 3.0 M solution in diethyl ether, 1.5 mmol) was added dropwise. The reaction mixture was allowed to warm to room temperature and stirred for 1 h, then cooled to 0 °C, and degassed water (0.5 mL) was added cautiously. Volatiles were removed *in vacuo*, and the resulting oil was redissolved in hexane (40 mL). The resulting suspension was filtered to remove insoluble impurities, and volatiles were removed *in vacuo* to afford **4** as a pale-yellow oil (0.41 g, 0.85 mmol, 86%).

**^1^H NMR** *δ*_H_ (400 MHz, CDCl_3_) 7.67–7.59 (4H, m, ArH-2, ArH-8, *m*-Ph CH), 7.32 (1H, d, ^3^*J*_HH_ = 7.1 Hz, ArH-7), 7.29–7.25 (3H, m, *o*/*p*-Ph CH), 7.16 (1H, d, ^3^*J*_HH_ = 7.1 Hz, ArH-3), 3.38 (4H, s, H-11, H-12, 2 × CH_2_), 2.25 (1H, septd, ^3^*J*_HH_ = 6.9 Hz, ^2^*J*_HP_ = 5.1 Hz, *i*Pr CH), 2.09 (1H, septd, ^3^*J*_HH_ = 7.0 Hz, ^2^*J*_HP_ = 3.2 Hz, *i*Pr CH), 1.22 (3H, dd, ^3^*J*_HP_ = 15.0 Hz, ^3^*J*_HH_ = 6.9 Hz, *i*Pr CH_3_), 1.18 (3H, d, ^6^*J*_HP_ = 1.8 Hz, Me(Sb)), 1.04 (3H, dd, ^3^*J*_HP_ = 14.7 Hz, ^3^*J*_HH_ = 6.9 Hz, *i*Pr CH_3_), 0.97 (3H, dd, ^3^*J*_HP_ = 12.5 Hz, ^3^*J*_HH_ = 7.0 Hz, *i*Pr CH_3_), 0.66 (3H, dd, ^3^*J*_HP_ = 11.8 Hz, ^3^*J*_HH_ = 7.0 Hz, *i*Pr CH_3_).

**^13^C{^1^H} NMR** *δ*_C_ (75 MHz, CDCl_3_) 149.1 (s, qC-6), 147.2 (d, ^4^*J*_CP_ = 1.6 Hz, qC-4), 146.0 (d, ^1^*J*_CP_ = 40.5 Hz, qC-9), 142.1 (s, qC-1), 140.0 (d, ^3^*J*_CP_ = 7.8 Hz, qC-5), 138.2 (s, C-2), 136.6 (s, *m*-Ph CH), 134.0 (s, qC-10), 133.9 (d, ^2^*J*_CP_ = 2.4 Hz, C-8), 130.1 (d, ^5^*J*_CP_ = 16.2 Hz, *i*-Ph qC), 128.4 (s, *o*-Ph CH), 127.6 (s, *p*-Ph CH), 120.0 (s, C-3), 119.0 (s, C-7), 30.3 (s, C-11/C-12), 30.0 (s, C-11/C-12), 26.1 (d, ^1^*J*_CP_ = 12.5 Hz, *i*Pr CH), 25.9 (d, ^1^*J*_CP_ = 13.7 Hz, *i*Pr CH), 20.6 (d, ^2^*J*_CP_ = 17.7 Hz, *i*Pr CH_3_), 20.1 (d, ^2^*J*_CP_ = 13.3 Hz, *i*Pr CH_3_), 20.0 (d, ^2^*J*_CP_ = 7.3 Hz, *i*Pr CH_3_), 19.2 (d, ^2^*J*_CP_ = 7.2 Hz, *i*Pr CH_3_), 4.74 (d, ^5^*J*_CP_ = 34.1 Hz, Me(Sb)).

**^31^P NMR** *δ*_P_ (109 MHz, CDCl_3_) −19.7 (m).

**^31^P{^1^H} NMR** *δ*_P_ (109 MHz, CDCl_3_) −19.6 (s).

**MS** (APCI+) *m*/*z* 390.05 (85%, M − Ph − Me), 405.07 (100, M − Ph), 467.09 (73, M − Me), 483.12 (11, M + H), 499.11 (4, M + O + H).

**HRMS** (APCI+) C_25_H_31_PSb (M + H)^+^; calculated: 483.1196; found: 483.1195.

### 3.6. [iPr_2_P-Ace-Sb(Ph)Me]Mo(CO)_4_, ***4.Mo(CO)_4_***

A solution of compound **4** (240 mg, 0.497 mmol) in dichloromethane (40 mL) was added to a stirred suspension of *cis*-tetracarbonyl(norbornadiene)molybdenum(0) (0.164 g, 0.546 mmol) in dichloromethane (10 mL), and the resulting suspension was stirred at room temperature for 3 days. Insoluble material was removed by filtration through celite, and volatiles were removed *in vacuo* to afford **4.Mo(CO)_4_** as a pale brown solid (329 mg, 0.476 mmol, 96%). Crystals suitable for single crystal X-ray diffraction were grown from hexane at −25 °C.

**^1^H NMR** *δ*_H_ (400 MHz, CDCl_3_) 7.79 (1H, d, ^3^*J*_HH_ = 6.9 Hz, ArH-2), 7.70 (1H, ≈ t, ^3^*J*_HP_ = 7.8 Hz, ^3^*J*_HH_ = 7.8 Hz, ArH-8), 7.56–7.50 (2H, m, *m*-Ph CH), 7.39–7.33 (4H, m, ArH-7, *o*/*p*-Ph CH), 7.30 (1H, d, ^3^*J*_HH_ = 6.9 Hz, ArH-3), 3.40 (4H, s, H-11/H-12), 2.50–2.38 (1H, m, *i*Pr CH), 2.35–2.25 (1H, m, *i*Pr CH), 1.58 (s, 3H, Me(Sb)), 1.23 (3H, dd, ^3^*J*_PH_ = 15.4 Hz, ^3^*J*_HH_ = 6.9 Hz, *i*Pr CH_3_), 1.10 (3H, dd, ^3^*J*_PH_ = 15.8 Hz, ^3^*J*_HH_ = 7.1 Hz, *i*Pr CH_3_), 1.04 (3H, dd, ^3^*J*_PH_ = 15.6 Hz, ^3^*J*_HH_ = 7.1 Hz, *i*Pr CH_3_), 0.99 (3H, dd, ^3^*J*_PH_ = 15.0 Hz, ^3^*J*_HH_ = 6.9 Hz, *i*Pr CH_3_).

**^13^C{^1^H} NMR** *δ*_C_ (101 MHz, CDCl_3_) 218.4 (d, ^2^*J*_CP_ = 8.1 Hz, CO), 216.4 (d, ^2^*J*_CP_ = 20.3 Hz, CO), 211.1 (d, ^2^*J*_CP_ = 9.3 Hz, CO), 210.8 (d, ^2^*J*_CP_ = 9.1 Hz, CO), 150.6 (s, qC-6), 150.4 (d, ^4^*J*_CP_ = 1.4 Hz, qC-4), 140.9 (d, ^3^*J*_CP_ = 6.5 Hz, qC-5), 139.5 (d, ^2^*J*_CP_ = 10.3 Hz, qC-10), 138.3 (s, C-2), 135.6 (d, ^3^*J*_CP_ = 2.3 Hz, *i*-Ph qC), 133.9 (s, *m*-Ph CH), 133.3 (s, C-8), 129.4 (s, *p*-Ph CH), 129.0 (s, *o*-Ph CH), 124.6 (d, ^1^*J*_CP_ = 20.4, qC-9), 123.4 (d, ^3^*J*_CP_ = 2.4 Hz, qC-1), 119.8 (s, C-3), 118.9 (d, ^3^*J*_CP_ = 5.7 Hz, C-7), 30.2 (s, C-11/C-12), 29.5 (s, C-11/C-12), 28.7 (d, ^1^*J*_CP_ = 15.9 Hz, *i*Pr CH), 28.1 (d, ^1^*J*_CP_ = 14.9 Hz, *i*Pr CH), 19.1 (d, ^2^*J*_CP_ = 4.3 Hz, *i*Pr CH_3_), 19.0 (d, ^2^*J*_CP_ = 4.1 Hz, *i*Pr CH_3_), 18.24 (d, ^2^*J*_CP_ = 5.9 Hz, *i*Pr CH_3_), 18.15 (d, ^2^*J*_CP_ = 4.4 Hz, *i*Pr CH_3_), 4.4 (d, ^3^*J*_CP_ = 2.9 Hz, Me(Sb)).

**^31^P NMR** *δ*_P_ (109 MHz, CDCl_3_) 43.3 (m).

**^31^P{^1^H} NMR** *δ*_P_ (109 MHz, CDCl_3_) 43.4 (s).

**IR** (KBr disk, cm^−1^) *ν*_max_ 3007 (*ν*_Ar–H_, w), 2963 (*ν*_C–H_, m), 2014 (*ν*_C≡O_, vs), 1899 (*ν*_C≡O_, vs), 1603 (m), 1261 (m), 1084 (s), 1023 (s), 802 (m), 733 (m), 694 (m), 613 (m), 585 (m), 453 (m).

**MS** (APCI+) *m*/*z* 271.16 (100%, M − Mo(CO)_4_ − Sb(Me)Ph + H), 287.16 (M − Mo(CO)_4_ − Sb(Me)Ph + O + H), 637.02 (1, M − 2CO + H)

**HRMS** (APCI+) C_27_H_31_MoO_2_PSb (M − 2CO + H)^+^; calculated: 637.0149; found: 637.0148.

### 3.7. iPr_2_P-Ace-Sb(Ph)iPr, ***5***

To a stirred suspension of **3** (1.00 g, 1.98 mmol) in tetrahydrofuran (20 mL), cooled to −78 °C, a solution of isopropylmagnesium chloride (1.5 mL, 1.70 M solution in THF, 2.55 mmol) was added dropwise. The reaction mixture was allowed to warm to room temperature with stirring overnight, then cooled to 0 °C, and degassed water (0.5 mL) was added cautiously. Volatiles were removed *in vacuo* to give an oil. Hexane (50 mL) was added, and the resultant suspension was filtered to remove the insoluble salts. Volatiles were removed *in vacuo* to afford **5** as a yellow oil, which crystallised to a yellow solid on standing at room temperature for several days (0.824 g, 1.61 mmol, 81%). Crystals suitable for single crystal X-ray diffraction were grown from ethanol at −25 °C. M. p. 73–76 °C.

Elemental Analysis: C_27_H_34_PSb; calculated (%) C 63.43, H 6.70; found (%) C 63.31, H 6.73.

**^1^H NMR** *δ*_H_ (500 MHz; CDCl_3_) 7.85 (1H, d, ^3^*J*_HH_ = 7.0 Hz, ArH-2), 7.61 (1H, dd, ^3^*J*_HH_ = 7.1 Hz, ^3^*J*_HP_ = 3.6 Hz, ArH-8), 7.58–7.54 (2H, m, *m*-Ph CH), 7.30 (1H, d, ^3^*J*_HH_ = 7.1 Hz, ArH-3), 7.26 (1H, d, ^3^*J*_HH_ = 7.1 Hz, ArH-7), 7.23–7.19 (3H, m, *o/p*-Ph CH), 3.39 (4H, s, H-11, H-12), 2.32 (1H, sept, ^3^*J*_HH_ = 7.2 Hz, *i*Pr(Sb) CH), 2.22 (1H, septd, ^3^*J*_HH_ = 7.1 Hz, ^2^*J*_HP_ = 3.4 Hz, *i*Pr(P) CH), 2.04–1.93 (1H, m, *i*Pr(P) CH), 1.36 (3H, d, ^3^*J*_HH_ = 7.2 Hz, *i*Pr(Sb) CH_3_), 1.25–1.19 (6H, m, *i*Pr(Sb) CH_3_, *i*Pr(P) CH_3_), 0.99 (3H, dd, ^3^*J*_HP_ = 11.9 Hz, ^3^*J*_HH_ = 7.0 Hz, *i*Pr(P) CH_3_), 0.95 (3H, dd, ^3^*J*_HP_ = 14.6 Hz, ^3^*J*_HH_ = 6.9 Hz, *i*Pr(P) CH_3_), 0.42 (3H, dd, ^3^*J*_HP_ = 12.4 Hz, ^3^*J*_HH_ = 7.0 Hz, *i*Pr(P) CH_3_).

**^13^C{^1^H} NMR** *δ*_C_ (101 MHz; CDCl_3_) 148.9 (s, qC-6), 147.3 (d, ^4^*J*_CP_ = 1.7 Hz, qC-4), 144.2 (d, ^1^*J*_CP_ = 28.2 Hz, qC-9), 142.2 (d, ^2^*J*_CP_ = 27.2 Hz, qC-10), 140.1 (d, ^3^*J*_CP_ = 7.8 Hz, qC-5), 138.1 (s, C-2), 136.7 (s, *m*-Ph CH), 134.0 (d, ^2^*J*_CP_ = 2.4 Hz, C-8), 133.9 (d, ^3^*J*_CP_ = 5.6 Hz, qC-1), 130.7 (d, ^5^*J*_CP_ = 17.3 Hz, *i*-Ph qC), 128.2 (s, *o*-Ph CH), 127.4 (s, *p*-Ph CH), 120.1 (s, C-7), 119.0 (s, C-3), 30.2 (s, C-11/C-12), 30.0 (s, C-11/C-12), 26.4 (d, ^1^*J*_CP_ = 14.8 Hz, *i*Pr(P) CH), 26.2 (d, ^1^*J*_CP_ = 14.0 Hz, *i*Pr(P) CH), 25.1 (d, ^5^*J*_CP_ = 36.8 Hz, *i*Pr(Sb) CH), 22.6 (s, *i*Pr(Sb) CH_3_), 21.7 (s, *i*Pr(Sb) CH_3_), 20.7 (d, ^2^*J*_CP_ = 16.8 Hz, *i*Pr(P) CH_3_), 20.1 (d, ^2^*J*_CP_ = 8.1 Hz, *i*Pr CH_3_), 19.9 (d, ^2^*J*_CP_ = 16.6 Hz, *i*Pr(P) CH_3_), 19.2 (d, ^2^*J*_CP_ = 9.3 Hz, *i*Pr(P) CH_3_).

**^31^P NMR** *δ*_P_ (109 MHz; CDCl_3_) −20.8 (m).

**^31^P{^1^H} NMR** *δ*_P_ (109 MHz; CDCl_3_) −20.8 (s).

**Raman:** (glass capillary, cm^−1^) *ν*_max_ 3038s (*ν*_Ar–H_), 2921vs (*ν*_C–H_), 1601s, 1562s, 1442s, 1325vs, 1001vs, 657m, 582s, 490s.

**MS** (APCI+) *m*/*z* 390.05 (100%, M − *i*Pr − Ph), 433.10 (60, M − Ph), 467.09 (50, M − *i*Pr), 511.15 (16, M + H), 527.15 (4, M + O + H).

**HRMS** (APCI+): C_27_H_35_PSb (M + H)^+^; calculated: 511.1509; found: 511.1510.

### 3.8. iPr_2_P-Ace-SbMe_2_, ***7***

A solution of **6** (0.49 g, 1.06 mmol) in diethyl ether (40 mL) was cooled to −78 °C. A solution of methylmagnesium bromide in tetrahydrofuran (0.7 mL, 3.0 M solution, 2.10 mmol, diluted with diethyl ether, 4 mL) was added dropwise with stirring over one hour. The resulting suspension was stirred at -78 °C for a further 90 min before warming to ambient temperature overnight. The suspension was filtered, and volatiles were removed from the filtrate *in vacuo,* affording **7** as an off-white solid (0.42 g, 94%). The yield is approximate, as ^1^H NMR spectra indicate the presence of solvated diethylether. M.p. 239 °C with decomposition.

**^1^H NMR:** δ_H_ (400.1 MHz, CDCl_3_) 7.77 (1H, d, ^3^*J*_HH_ 7.0 Hz, H-2), 7.66 (1H, dd, ^3^*J*_HH_ 7.1, ^3^*J*_HP_ 3.6 Hz, H-8), 7.32 (1H, d, ^3^*J*_HH_ 7.1 Hz, H-7), 7.26 (1H, d, ^3^*J*_HH_ 7.0 Hz, H-3), 3.38 (4H, s, H-11, 12), 2.19 (2H, br sept, ^3^*J*_HH_ 7.0 Hz, *i*Pr CH), 1.17 (6H, dd, ^3^*J*_HP_ 14.6, 3*J*_HH_ 6.9 Hz, *i*Pr CH_3_), 0.96 (6H, d, ^6ts^*J*_HP_ 1.2 Hz, Sb-CH_3_), 0.94 (6H, dd, ^3^*J*_HP_ 12.3, ^3^*J*_HH_ 7.0 Hz, *i*Pr CH_3_).

**^13^C DEPTQ NMR:** δ_C_ (100.6 MHz, CDCl_3_) 149.0 (s, qC-6), 147.0 (s, qC-4), 141.5 (d, ^2^*J*_CP_ 4.3 Hz, qC-10), 139.8 (d, ^3^*J*_CP_ 1.7 Hz, qC-5), 136.0 (s, C-2), 133.8 (d, ^2^*J*_CP_ 2.5 Hz, C-8), 133.3 (d, ^3^*J*_CP_ 5.1 Hz, qC-1), 130.3 (d, ^1^*J*_CP_ 20.0 Hz, qC-9), 119.9 (s, C-3), 118.9 (s, C-7), 30.2 (s, C-11/12), 29.9 (s, C-11/12), 26.0 (d, ^1^*J*_CP_ 13.5 Hz, *i*Pr CH), 20.4 (d, ^2^*J*_CP_ 16.7 Hz, *i*Pr CH_3_), 19.9 (d, ^2^*J*_CP_ 9.0 Hz, *i*Pr CH_3_), 3.4 (d, ^5ts^*J*_CP_ 34.9 Hz, Sb-CH_3_).

**^31^P{^1^H} NMR:** δ_P_ (162.0 MHz, CDCl_3_) −18.5 (s).

**Raman:** (glass capillary, cm^−1^) *ν*_max_ 2933vs (ν_C-H_), 2124w, 1601m, 1562m, 1447m, 1325vs, 577s, 509s, 482vs.

**HRMS** (ASAP+): *m*/*z* Calcd. for C_20_H_29_PSb 421.1045, found 421.1042 [M + H]; Calcd. for C_19_H_25_PSb 405.0732, found 405.0724 [M − Me].

### 3.9. iPr_2_P-Ace-Sb(nBu)_2_, ***8***

A solution of **6** (1.00 g, 2.16 mmol) in diethyl ether (40 mL) was cooled to −78 °C. To this, a solution of n-butyllithium in hexane (1.7 mL, 2.5 M solution, 4.25 mmol) was added dropwise with stirring over one hour. The resulting suspension was stirred at this temperature for a further hour before warming to ambient temperature overnight. The suspension was filtered, and the volatiles were removed from the filtrate *in vacuo*, affording **8** as a yellow oil (yield quantitative). A few crystals suitable for single-crystal X-ray diffraction formed spontaneously from the oil after long standing at room temperature. M. p. 53 °C.

**^1^H NMR**: δ_H_ (400.1 MHz, CDCl_3_) 7.73 (1H, d, ^3^*J*_HH_ 7.0 Hz, H-2), 7.67 (1H, dd, ^3^*J*_HH_ 7.1, ^3^*J*_HP_ 3.6 Hz, C-8), 7.31 (1H, d, ^3^*J*_HH_ 7.1 Hz, H-7), 7.25 (1H, d, ^3^*J*_HH_ 7.0 Hz, H-3), 3.38 (4H, s, H-11, 12), 2.22 (2H, dsept, ^3^*J*_HH_ 6.9, ^2^*J*_HP_ 3.9 Hz, *i*Pr CH), 1.71−1.48 (8H, m, SbC*H*_2_ and SbCH_2_C*H*_2_), 1.44−1.36 (4H, m, Sb CH_2_CH_2_C*H*_2_), 1.23 (6H, dd, ^3^*J*_HP_ 14.4, ^3^*J*_HH_ 6.9 Hz, 2× *i*Pr CH_3_), 0.95 (6H, dd, ^3^*J*_HP_ 12.4, ^3^*J*_HH_ 7.2 Hz, 2× *i*Pr CH_3_), 0.90 (6H, t, ^3^J_HH_ 7.2 Hz, 2× n-Bu CH_3_).

**^13^C DEPTQ NMR**: δ_C_ (100.6 MHz, CDCl_3_) 148.9 (s, qC-6), 146.8 (d, ^4^*J*_CP_ 1.7 Hz, qC-4), 142.1 (d, ^2^*J*_CP_ 26.9 Hz, qC-10), 139.9 (d, ^3^*J*_CP_ 7.8 Hz, qC-5), 136.5 (s, C-2), 133.8 (d, ^2^*J*_CP_ 2.4 Hz, C-8), 131.9 (d, ^3^*J*_CP_ 5.0 Hz, qC-1), 130.9 (d, ^1^*J*_CP_ 18.0 Hz, qC-9), 119.9 (s, C-3), 118.9 (s, C-7), 30.2 (s, SbCH_2_*C*H_2_), 29.9 (s, C-11, 12), 27.0 (s, SbCH_2_CH_2_*C*H_2_), 26.4 (d, ^1^*J*_CP_ 14.3 Hz, *i*Pr CH), 20.7 (d, ^5ts^*J*_CP_ 30.6 Hz, Sb*C*H_2_), 20.5 (d, ^2^*J*_CP_ 16.9 Hz, *i*Pr CH_3_), 20.2 (d, ^2^*J*_CP_ 9.5 Hz, *i*Pr CH_3_), 14.0 (s, n-Bu CH_3_).

**^31^P{^1^H} NMR**: δ_P_ (162.0 MHz, CDCl_3_) −18.5 (s)

**Raman**: (glass capillary, cm^−1^) *ν*_max_ 2919vs (ν_C−H_), 2868vs, 1556m, 1435m, 1325s, 1157w, 583m.

**HRMS** (ASAP+): *m*/*z* Calcd. for C_26_H_41_PSb [M + H]: 505.1984, found 505.1986.

### 3.10. X-ray Diffraction

X-ray diffraction data for compound **2.PtCl2** were collected at 125 K using the St Andrews Automated Robotic Diffractometer (STANDARD) [43], consisting of a Rigaku sealed-tube X-ray generator equipped with a SHINE monochromator [Mo Kα radiation (λ = 0.71075 Å)], and a Saturn 724 CCD area detector, coupled with a Microglide goniometer head and an ACTOR SM robotic sample changer. Diffraction data for compounds **4.Mo(CO)_4_**, **5** and **8** were collected at 173 K using a Rigaku FR-X Ultrahigh Brilliance Microfocus RA generator/confocal optics [Mo Kα radiation (λ = 0.71075 Å)] with an XtaLAB P200 diffractometer. Intensity data for all compounds were collected using ω steps, accumulating area detector images spanning at least a hemisphere of reciprocal space. Data for all compounds analysed were collected using CrystalClear [44] and processed (including correction for Lorentz, polarization and absorption) using either CrystalClear or CrysAlisPro [45]. Structures were solved by direct (SHELXS [46]), Pattterson (PATTY [47]) or charge-flipping (Superflip [48]) methods and refined by full-matrix least-squares against F^2^ (SHELXL-2019/3 [49]). Non-hydrogen atoms were refined anisotropically, and hydrogen atoms were refined using a riding model. In **5**, both isopropyl groups bound to phosphorus were disordered over two positions. Atoms were split and refined with partial occupancies, and restraints to bond distances were required. Crystals of **8** were affected by pseudo-merohedral twinning, showing a twin law of [−0.9999 0.0198 0.0009 −0.0032 0.9992 −0.0203 −0.0007 −0.1315 −0.9981] and a refined twin fraction of 0.489. All calculations were performed using the CrystalStructure interface [50]. Selected crystallographic data are presented in Table 4.

### 3.11. Computational Methodology

#### 3.11.1. Geometry Optimisations and QTAIM Analysis

Geometry optimisations were performed for models **5** and **8** using coordinates derived from their X-ray crystal structures. These models were geometry optimised without restraints using the ORCA 5.0.4 software package [51] utilising the PBE0 density functional [52] and all-electron ZORA corrected [53] def2-TZVP basis sets [54,55,56,57] for all atoms (except Sb), SARC-ZORA-TZVP [58] basis sets for the Sb atoms, along with SARC/J auxiliary basis sets decontracted def2/J up to Kr [59] and SARC auxiliary basis sets beyond Kr. [58,60,61,62]. Gradient corrections were performed with Grimme’s 3rd generation dispersion correction [63,64]. TightSCF and TightOpt convergence criteria were employed, and the location of true minima in these optimisations was confirmed by frequency analysis, which demonstrated that no imaginary vibrations were present.

Extended wavefunction (.wfx) files were generated from these optimisations using Orca 5.0.4 [51]. AIM analysis was performed using MultiWFN 3.8 [65].

#### 3.11.2. EDA-NOCV Analysis

EDA-NOCV calculations were carried out on models of **5** and **8** using coordinates derived from the geometry-optimised structures. The compounds were divided into an Acenap(P*i*Pr_2_)^−^ and SbR_2_^+^ fragments (**5** = ^+^Sb(*i*Pr)Ph; **8** = ^+^Sb(*n*Bu)_2_). Calculations were carried out using the ORCA 5.0.4 software package [51] utilising the PBE0 density functional [52,66] and all-electron ZORA corrected [53] def2-TZVP basis sets [54,55,56,57] for all atoms (except Sb), SARC-ZORA-TZVP basis sets [58] for the Sb atoms, along with SARC/J auxiliary basis sets decontracted def2/J up to Kr [59] and SARC auxiliary basis sets beyond Kr [58,60,61,62]. Gradient corrections were performed with Grimme’s 3rd generation dispersion correction [63,64]. VeryTightSCF convergence settings and an integration accuracy value of 6.0 were employed. Deformation density plots were calculated from .cube files of the relevant NOCVs using the MultiWFN 3.8 software package [65] and visualised using Avogadro v1.2.0 [67].

## 4. Conclusions

The synthetic utility of *peri*-substituted phosphine-chlorostibines **3** and **6** in reactions with carbon nucleophiles has been demonstrated. Reactions with Grignard reagents or *n*BuLi proceeded rather cleanly and gave alkyl/aryl and alkyl tertiary stibines **4**, **5**, **7** and **8** with very good yields.

The coordination chemistry of the selected tertiary phosphine-stibines has also been probed. Two complexes, **2.PtCl_2_** and **4.Mo(CO)_4_**, have been synthesised. Single-crystal X-ray diffraction confirmed that both the phosphine and the stibine groups are attached to the platinum(II) and Mo(0) centres.

In the phosphine-phosphine *peri*-substituted species, such as *i*Pr_2_P-Ace-PPh_2_, large magnitude ^4TS^*J*_PP_ (180 Hz for the above species) were observed due to the forced overlap of the two lone pairs on the phosphorus atoms [68]. As antimony has no spin ½ isotopes, the direct observation of P−Sb couplings (formally ^4TS^*J*_PSb_) was not possible in the phosphine-stibines reported here. However, long-range ^5TS^*J*_CP_ couplings of up to 36.8 Hz were observed for carbon atoms attached to Sb atoms in all the phosphine stibines. This indicates the presence of a strong through-space coupling pathway through the phosphorus and antimony atoms (both possessing a lone pair), and the through-bond pathway contribution (^5^*J*) is expected to be negligible [69].

The QTAIM analysis supports the existence of a P∙∙∙Sb interaction in **5** and **8,** which is not purely closed-shell (i.e., Van der Waals), and the visualisation of the deformation densities in an EDA-NOCV analysis supports the view that electron density from the P atom flows towards an apparent Sb–C σ*orbital.

## Data Availability

CCDC 2337824-2337827 contains the supplementary crystallographic data for this paper. These data can be obtained free of charge from The Cambridge Crystallographic Data Centre via www.ccdc.cam.ac.uk/structures. The research data underpinning this publication can be accessed at https://doi.org/10.17630/8f5cd85b-0e47-44ab-9d9a-e27d44a123f6 [70].

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
