# Peer review of "Synthesis and Structural Studies of peri-Substituted Acenaphthenes with Tertiary Phosphine and Stibine Groups†"

_molecules, 2024, doi:10.3390/molecules29081841_

Round 1

Reviewer 1 Report

Comments and Suggestions for Authors

The manuscript molecules-2951699 is an excellent text covering the synthesis and structural studies of peri-substituted 2 acenaphthenes with tertiary phosphine and stibine groups (as stated in the title). The manuscript is well organised and fluent and should be accepted in its present form. The only corrections needed in the text are in the references. Error! Reference source not found occurs throughout the text. Other corrections relate to supplementary material: Authors should include raw data for NMR, IR, Raman and MS spectra.

Comments on the Quality of English Language

English language is appropriate. 

Reviewer 2 Report

Comments and Suggestions for Authors

This work is concentrated on the reaction of peri-substituted phosphine-chlorostibines with carbon nucleophiles such as Grignard reagents or nBuLi. The preparation and characterization of peri-substituted phosphine-stibines were described. The coordination of phosphine-stibines with PtCl2 and Mo(CO)4 give complexes 2.PtCl2 and 4.Mo(CO)4. X-ray crystallography confirmed that both the phosphine and the stibine groups are bonded to the platinum(II) and Mo(0) centres. Theoretical calculation study supports the existence of a P∙∙∙Sb interaction, which is not purely closed-shell (i.e. Van der Waals) owing to mostly electron density flowing from the P atom towards apparent Sb–C σ*orbital.

1.      Both 7 and 8 decomposed in chloroform solutions within several days, indicating instability in halogenated solvents. Why ? Does peri-substituted phosphine-stibine react with chloroform ?

2.      Are platinum(II) and Mo(0) complexes of peri-substituted phosphine-stibines sensitive to air or water ? How about their thermal stability ?

3.      It is necessary to investigate the UV-Vis absorption and emission properties of peri-substituted phosphine-stibines as well as their corresponding platinum(II) and Mo(0) complexes. Getting insight on transition character in UV-Vis absorption spectra is also necessary.

4.   There are a number of “Error! Reference source not found”.

Comments on the Quality of English Language

Moderate editing of English language is required.

Reviewer 3 Report

Comments and Suggestions for Authors

Manuscript presented by Laurence J. Taylor et al. shows study about synthesis and structural aspects of peri-substituted acenaphthenes with tertiary phosphine and stibine groups.

After analyzing the presented document, in my humble opinion it can be published, but after a major revision. Comments containing changes for consideration:

(1)   In title and abstract section should underline if the obtained compounds are new connections.

(2)   I suggest to add “QTAIM” and EDA-NOCV” to keywords.

(3)   The manuscript requires a thorough editorial revision. In some places the template is used incorrectly (see citation style). What is more, some references are added in not correct way.

(4)   Please present your work motivation in detail. Please support your background with appropriate literature. The sentence "Tertiary amines and phosphines play a key role as tuneable ligands, with uses in transition metal catalysis and other applications" is insufficient.

(5)   Figure 1 - please describe in detail the meaning of the presented compounds.

(6)   Please indicate how the reaction monitoring was carried out and provide details (solvent ect).

(7)   Please check that all Tables, Figures, Schemes etc are cited in the main text.

(8)   Please justify the chosen technique for quantum calculation.

(9)   I do not see supporting information in the materials submitted for evaluation. Please prepare a separate file containing all spectra for the presented compounds. Please consider moving all metrics of compound to this file.

(10)  In the conclusion part, please focus more on synthetic aspects. Please add information abot how many compounds were synthesized and which were novel connections.

Round 2

Reviewer 3 Report

Comments and Suggestions for Authors

The manuscript has been improved. I recommend its publication in Molecules in a current form.